# An Evidence-Based Narrative Review of Scleral Hypoxia Theory in Myopia: From Mechanisms to Treatments

**DOI:** 10.3390/ijms26010332

**Published:** 2025-01-02

**Authors:** Qin Xiao, Xiang Zhang, Zhang-Lin Chen, Yun-Yi Zou, Chang-Fa Tang

**Affiliations:** 1College of Physical Education, Hunan Normal University, Changsha 410012, China; qinzixiao@hnfnu.edu.cn (Q.X.); zhangx581@hunnu.edu.cn (X.Z.); zhanglinchen@hunnu.edu.cn (Z.-L.C.); 2College of Physical Education, Hunan First Normal University, Changsha 410205, China

**Keywords:** myopia, hypoxia, mechanisms, treatments

## Abstract

Myopia is one of the dominant causes of visual impairment in the world. Pathological myopia could even lead to other serious eye diseases. Researchers have reached a consensus that myopia could be caused by both environmental and genetic risk factors. Exploring the pathological mechanism of myopia can provide a scientific basis for developing measures to delay the progression of myopia or even treat it. Recent advances highlight that scleral hypoxia could be an important factor in promoting myopia. In this review, we summarized the role of scleral hypoxia in the pathology of myopia and also provided interventions for myopia that target scleral hypoxia directly or indirectly. We hope this review will aid in the development of novel therapeutic strategies and drugs for myopia.

## 1. Introduction

Our eyes are a precisely regulated optical system. We can see objects clearly because the light emitted or reflected by them is focused on the retina (the light-sensitive part of the eye). If the light is not properly focused on the retina, the resulting image is blurry, which is known as a refractive error. In a relaxed state of accommodation, parallel light enters the eye and focuses in front of the retina, leading to myopia. Myopia is considered as an eye disease. It is a type of refractive error which mainly manifests as blurred vision when looking at distant objects. Myopia is one of the leading causes of visual impairment in the world. The prevalence of myopia has significantly increased over the last two decades. In 2000, 1.46 billion people were myopic, accounting for 22.9% of the world’s population. In 2000, 163 million people, or 2.7% of the world’s population, had high myopia of −6.00 diopters or worse. By 2050, the global prevalence of myopia is expected to reach 49.8%, with 9.8% classified as high myopia [1]. High myopia could finally lead to pathologic myopia, which is characterized by posterior staphyloma, diffuse choroidal atrophy, and retinal complication [2,3]. The occurrence of myopia is affected by multiple factors, such as genetics, the environment, and bad eye habits. Researchers have found that environmental factors are the main reasons for the surge in the incidence of myopia, especially long hours of near work (with eyes) [4]. In recent years, the prevention and control of myopia has become a difficult issue and a hot topic globally. In China, the incidence of myopia in children and adolescents is high and increases progressively year by year, which seriously impacts the physical and mental health of children. In 2018, myopia prevention and control were elevated to a national strategy in China. During the COVID-19 pandemic, the incidence of myopia rebounded significantly, making its prevention and control an urgent issue. Exploring the pathogenesis of myopia is essential for its prevention and treatment.

Previous studies on the pathological mechanism of myopia have focused on ocular inflammation [5], dopamine (neurotransmitter) signaling [6], nitric oxide [7], TGF-β [7], etc. Despite the large amount of myopia research, the molecular/cellular mechanisms underlying myopia development are poorly understood, hindering the search for the most effective pharmacological control. The anticholinergic blocker atropine is by far the most effective medication for myopia and is used by clinicians in an off-label manner for myopia control. Although the exact mechanism of action of atropine remains elusive and controversial, the complex interaction of atropine with receptors on different ocular tissues at multiple levels can be classified as a shotgun approach to myopia [8]. Thus, exploring the clear pathological mechanism of myopia can provide more targets for the prevention and treatment of myopia.

Hypoxia-inducible factors (HIFs) are a class of transcription factors that are highly sensitive to oxygen concentration [9]. The expression level of HIFs can increase in a hypoxic environment induced by high glucose, ischemia, and other conditions to regulate related molecular biological mechanisms [9,10]. Recently, researchers have found that scleral hypoxia is a contributing factor in myopia and could aggravate the occurrence and development of myopia [11,12,13]. But few studies have been concerned with scleral hypoxia and the underlying mechanism of how hypoxia induces myopia is still unclear.

This article reviews the progress of research on the role of hypoxia and HIFs in the pathogenesis of myopia. Furthermore, it also discusses potential anti-hypoxia measures and provides new ideas for the prevention and treatment of myopia.

## 2. Predisposing Factors of Pathological Myopia

Historically, some eye care professionals viewed myopia as a genetic abnormality, while others believed that it was caused by the environmental factors. However, human and animal studies conducted over the last four decades have shown that the development of myopia is influenced by both environmental and genetic factors [14,15]. Population studies have shown that environmental factors, such as prolonged near work and extensive reading, play an important role in the development of myopia [16,17]. In addition, many epidemiological studies have shown that myopia is more common in urban areas, among professionals, highly educated patients, computer users, and university students; it is associated with increased intelligence (aptitude) [18,19]. Furthermore, with the growing use of electronic devices such as mobile phones and computers, the prevalence of myopia is also rising, especially in children.

## 3. Structural Pathological Changes in Myopic Eyes

Ocular structural changes in myopia mainly involve changes in the morphology of the eyeball and intraocular tissues, and these changes are usually associated with an increase in axial length [3,20] (Figure 1). The characteristics of non-pathological ocular tissue changes in axial myopia are as follows: the elongation of the eye axis causes the eyeball to change from an oblate spherical or spherical to a long oval; the posterior equatorial part of the eyeball is the center of the expansion of the eyeball wall, and the horizontal diameter and vertical diameter of the eyeball increase slightly, except for the elongation in the axial direction. And in axial myopia [21], the decrease in photoreceptors, retinal pigment epithlium (RPE) cell density, and total retinal thickness are most significant in the posterior equatorial region of the eye, but not in the macular region at the posterior pole. The thinning of choroid and sclera is most obvious in the subfoveal region. Changes in the choroid mainly involve the large and middle vascular layers, and some high myopic eyes have the para-optic space in the choroid [22]. There is significant remodeling of the scleral tissue in myopic eyes, accompanied by changes in the extracellular matrix and scleral fibroblasts [23].

## 4. Scleral Remodeling in Myopia

The sclera is the outermost layer of the eyeball wall, which is thin but tough and elastic [24]. Recent years have seen significant advances in our knowledge of scleral biomechanics. The changes in its thickness and biological characteristics will directly affect the shape of the eyeball and the length of the axial length. Most cells in the sclera are fibroblasts, which secrete collagen and other extracellular matrix components. The extracellular matrix of the sclera is mainly composed of collagen type I, which supports fibroblasts and also determines the biomechanical properties of the sclera.

When myopia occurs, the sclera undergoes reshaping and its thickness becomes thinner, ultimately leading to its biomechanical properties changing [25]. Previous studies have shown that type I collagen content in the sclera begins to decrease and fibroblasts gradually differentiate when myopia occurs [26]. In detail, during this process, the retina-sclera signaling cascade promotes the increase in MMP-2, the degradation of TGF-β, and scleral myofibroblast transdifferentiation, resulting in abnormal extracellular matrix (ECM) metabolism and a reduction in collagen type I and glycosaminoglycans (GAGs). Subsequently, the sclera undergoes tissue remodeling and thinning of the posterior pole, which leads to axial myopia [5]. Despite significant global research interest, the specifics of the retina–choroid–scleral signaling pathways and the exact causes of these changes remain unclear [27] (Figure 2).

## 5. Hypoxia and Hypoxia-Inducible Factors in Ocular Diseases

### 5.1. Introduction of Hypoxia and Hypoxia-Inducible Factors

Oxygen (O2) is essential for cellular metabolism and biochemical reactions. Hypoxia occurs when demand for oxygen exceeds supply. Physiological hypoxia is beneficial and helps to maintain normal functional homeostasis, while pathological hypoxia is harmful because it aggravates inflammatory response and tissue dysfunction. Hypoxia has become one of the important features of many diseases, including liver and intestinal diseases [28], brain-related diseases [29], kidney disease [30], and cancer [31,32].

Hypoxia-inducible factors (HIFs) are crucial for activating adaptive and survival responses when suffering hypoxic stress. HIFs were discovered in 1990 as key transcription factors that regulate genes related to cellular energy metabolism, oxygen delivery, and cell survival [33]. HIFs belong to the PER-ARNT-SIM subfamily of the basic helix–loop–helix transcription factors. They are evolutionarily conserved in all metazoans [34]. In vertebrates, HIFs are mainly composed of three subunits of the α subtype (IF1α, HIF2α, and HIF3α) and one subunit of the β subtype (constitutive) (HIF1β, arylhydrocarbon receptor nuclear translocator (Arnt)). Under normoxic conditions, the specific proline residues on HIFα are hydroxylated by HIF prolyl hydroxylase domain-containing proteins (HIF-PHDs), and subsequently HIFα is degraded by the proteasome [35]. This process involves the binding and initiation of ubiquitination of von Hippel–Lindau (pVHL) and is facilitated by a complex including extensin B, extensin C (EloBC), cullin2 (CUL2), and loop box protein 1 (RBX1). Additional E3 ubiquitin ligases are also involved in this process, which ultimately leads to proteasomal degradation of HIFα81. Under hypoxic conditions, the activity of PHDs and FIH is reduced, allowing for the stabilization and accumulation of active HIF1β complexes, leading to subsequent binding to hypoxia response elements in target genes, with a concomitant induction of HIF target genes. HIFs, as transcription factors, will alter the expression of genes involved in glycolysis, angiogenesis, and cell survival, and more than 70 genes are known to be regulated by HIF-1 [33]. Therefore, HIFs and their downstream targets are gradually becoming new therapeutic options for the treatment of various organ injuries [36].

### 5.2. Role of Hypoxia Signaling in Ocular Diseases

A previous review highlighted the importance of hypoxia signaling, especially HIF-1, in the development and progression of various vision-threatening pathologies [37]. For instance, HIF-1 could promote the gene expression of fibrosis and neovascularization in the AMD retina [38,39] and regulate oxidative stress-induced cell death by sodium iodate and iron-dependent ferroptosis in AMD pathophysiology [40]. Researchers have summarized the underlying mechanism regulated by hypoxia signaling in ocular diseases, most of which were related to angiogenesis, fibrosis, and cell survival [37]. Detailed transactivation and target genes of HIF-1 can be found in this review [37]. Notably, some researchers have found that retinal/choroidal endothelial cells show notable heterogeneity in function, structure, and disease [41], especially in the response to hypoxia [42].

## 6. Studies on Hypoxia in Myopia

Considering the importance of hypoxia and HIFs in physiology and pathology, researchers have shifted their attention to exploring the crucial role of hypoxia and HIFs in myopia. Here, we summarize recent advances in this area (Table 1).

### 6.1. Scleral Hypoxia Is the Inducing Factor of Extracellular Matrix Remodeling and Myopia

In animal models of myopia [47] and in humans [48], myopia development is accompanied by the thinning of the sclera, the structural framework that maintains the shape and integrity of the eye. Scleral changes in different myopia models have prompted studies to investigate the mechanisms of scleral thinning and weakening during myopia progression. These studies suggest that myopia-related remodeling of the scleral extracellular matrix (ECM) is associated with a reduced synthesis and accelerated degradation of ECM components [49]. Remodeling of the scleral ECM weakens the scleral structural framework, thereby allowing for increased eye elongation. For example, a decrease in the expression of collagen type I, which is a major component of the scleral ECM, diminishes the scleral structural framework. During myopia development, the metabolism of type I collagen is increased due to the downregulation of its synthesis, as well as increased degradation [50,51]. Despite such insights and the many experimental models that have been developed, the physiological signals that trigger these changes remain elusive. Key questions remain as to how myopic scleral ECM remodeling is induced.

Previous studies have made it experimentally difficult to isolate and analyze each of the different scleral cell types to identify cell-specific signaling candidates that trigger scleral ECM remodeling and myopia development. Researchers have applied single-cell RNA sequencing (scRNA-seq) to resolve the complexity of different cell types in the retina [52]. Recently, in a study, researchers used scRNA-seq to reveal the different gene expression profiles of scleral cell populations and to explain the cell phenotypic changes (i.e., transdifferentiation from A1 fibroblasts to A2 myofibroblasts) and ECM changes in the sclera during myopia [11]. Using the Fluidigm C1 system, researchers determined the transcriptome of 93 single cells isolated from the sclera of a form deprivation (FD) model (untreated group as a control) and identified the presence of both A1 and A2 fibroblast populations in both FD and control eyes. The proportion of A2 cells in FD eyes was significantly higher than that in control eyes, and the scleral matrix genes Col1a1 and Col1a2 were significantly downregulated in A2 cells and the expression of myofibroblast transdifferentiation marker Acta2 was upregulated. Notably, scRNA-seq further showed that hypoxia signal (HIF-1α) is an important mechanism leading to scleral ECM remodeling [11]. Using protein–protein interaction (PPI) networks, further studies have found the presence of interactions between human myopia risk genes and genes involved in hypoxia [53]. Subsequently, in two myopia models of guinea pigs (FD and negative lens induction), scleral HIF-1α level was found to be increased in the myopic sclera and affected scleral ECM remodeling; in vitro experiments have also demonstrated that hypoxia induces myofibroblast transdifferentiation of human scleral fibroblasts [11]. In addition to HIF1a, the role of HIF2a in the sclera was found to be similar to that of HIF1a. Hypoxia also promoted the expression of HIF2a in visual deprivation mouse sclera and human scleral fibroblasts. Scleral HIF-2α knockdown significantly delayed myopia development in FD mice, with an inhibition of scleral MMP-2 upregulation and COL1α1 reduction [12].

In addition, gene set enrichment analysis (GSEA) of new genome-wide association study (GWAS) data of 593 patients with high myopia (diopter ≤ −6 diopter [D]) found that the HIF-1α signaling pathway was significantly enriched in patients with high myopia (diopter ≤ −10D) [13]. This suggests that HIF-1α signaling also contributes to human myopia by mediating interactions between genetic and environmental factors. In summary, these latest studies validate the important role of hypoxia in scleral ECM remodeling and myopia development and propose a therapeutic approach to control myopia by mitigating hypoxia.

### 6.2. Decreased Choroidal Blood Flow-Induced Scleral Hypoxia

The choroid is a highly vascularized structure in the posterior part of the uvea. Previous studies have shown that there is a correlation between choroidal thickness (chT) and myopia [54]. Choroidal thinning is a structural feature of myopia, and choroidal thickness is negatively correlated with axial length, suggesting that changes in choroidal thickness may be a predictive biomarker for long-term changes in eye elongation. In myopic people, choroidal thickness decreases by 13.62 μm for every degree of diopter increase [55]. Previous animal studies have also found that in chickens [56], guinea pigs [57], and monkeys [58], the choroid thinned during myopia development and thickened during recovery.

The choroid also regulates about 85% of outer retinal blood flow; choroidal blood perfusion (ChBP) is also the carrier of optical signals into biochemical signals during the homeostatic development of the eye. Given that the choroid is mainly a vascular structure that can rapidly change the blood flow in the eye, and changes in choroidal thickness may be mainly caused by the changes of choroidal blood flow [59]. Researchers have found that decreased choroidal blood flow is an important marker for the development of myopia [59,60]. Clinically, significant reductions in ChT and ChBP have been observed in patients with high myopia [61,62].

Choroidal blood flow is the main source of oxygen and nutrient supply, and reduced choroidal blood flow leads to hypoxia in ocular structures. It has been mentioned that scleral hypoxia is an important factor inducing myopia, and the fact that a reduction in choroidal blood flow is its upstream inducing factor has also been proven by recent studies [43,44]. Researchers reduced choroidal blood flow in the right eye of guinea pigs by cutting the temporociliary artery or injecting phenylephrine into the subperibulbaric space daily for a week. It was found that both methods can reduce choroidal blood flow, leading to scleral hypoxia and increased α-SMA expression, and induce the development of myopia. The use of quinirol, a drug that enhances form deprivation myopia, further reduces choroidal thickness and blood flow and increases scleral hypoxia, levels of α-SMA protein, and the degree of myopia [44].

From the point of view that choroidal thickness and blood flow can affect scleral hypoxia, this may provide a new predictor of myopia onset and a new target for treating myopia.

### 6.3. Retinal Dopamine Homeostasis Affects Choroidal Ischemia

Dopamine (DA) is an important neurotransmitter in the retina, which is produced and released by a subset of DAergic amacrine cells (DACs) that are sparsely distributed in the retina [63]. DA release and synthesis are light-dependent and can increase with successive diffuse and flickering light stimuli. It can activate D1 (D1-like receptors) and D2 (D2-like receptors) dopamine receptors distributed throughout the retina to mediate a variety of functions such as retinal development, visual signal and refractive development [64]. The effects of DA on retinal signaling are extensive and occur at multiple levels from photoreceptors to ganglion cells [65]. Therefore, it is important to explore the potential role of DA in retinal signaling in the development of myopia due to altered visual input.

DA and 3, 4-dihydroxyphenylacetic acid (DOPAC) levels have been found to be reduced in models of form deprivation and lens defocus in chickens, guinea pigs and monkeys [66,67,68,69]. In an experiment, intravitreal-injected atropine (250 µg or 360 nMol), atropine combined with the dopamine antagonist aspirin (500 µMol), or aspirin alone induced changes in retinal DA release and choroidal thickness in chickens tracked by optical coherence tomography (OCT), correlation analysis showed that the higher the retinal dopamine level release, the thicker the choroid [45]. As mentioned above, choroidal thickness has a certain effect on choroidal blood flow, which may eventually lead to scleral hypoxia.

DA has been previously considered a signal to prevent further development of myopia, and many studies have also shown that DA has anti-myopia effects. However, the exact mechanism of action and potential interactions with retinal pathways remain unclear. The finding that scleral hypoxia induces myopia may further demonstrate the role of DA in the development of myopia. Therefore, the discovery of the theory of scleral hypoxia also proposed that the retino-chorio-scleral hierarchy affects the development of myopia. The visual information that induces myopia, such as near work, can damage the retinal DA homeostasis, cause a reduction in choroidal blood flow, and then cause scleral hypoxia, leading to the remodeling of scleral extracellular matrix and finally to myopia.

### 6.4. Hypoxia-Induced Histone Lactylation Induces Transdifferentiation of Scleral Fibroblasts into Myofibroblasts and Promotes Myopia

Recent studies have shown that lactate-derived histone lactylation is involved in epigenetic regulation of many diseases [70,71]. Recent studies have also shown that hypoxic glycolytic reprogramming and histone emulsification in the sclera may be contributing factors to the transdifferentiation (FMT) of scleral fibroblasts into myofibroblasts during myopia [46]. Researchers found that the scleral glycolytic pathway was activated during myopia progression, and the inhibition of glycolysis or lactate production could inhibit FMT and myopia. Mechanistically, increased lactate induced H3K18 lactylation and subsequently promoted Notch1 expression, which induced the development of FMT and myopia, as revealed by H3K18la cuttag experiments.

The finding that glycolysis–lactate–histone lactylation enhances the myopic response provides new evidence and a mechanistic understanding for the theory of scleral hypoxia and also provides a new potential target for treating myopia by modulating scleral hypoxia.

## 7. Hypoxia Is a Target for the Treatment of Myopia

Scleral hypoxia plays an important role in the development of myopia and provides new clues for the interaction between ocular tissues. Scleral hypoxia theory implies that myopia can be inhibited by appropriate anti-hypoxia drugs or by improving the oxygen supply of other ocular structures (retina or choroid) to help prevent myopia.

### 7.1. ω-3 Polyunsaturated Fatty Acid (ω-3 PUFA) Dietary Supplements

Because of the cardiovascular health benefits of dietary supplementation with ω-3 polyunsaturated fatty acids (ω-3 PUFAs), the Food and Agriculture Organization of the United Nations (2010) recommends a daily intake of supplements. Specifically, the organization supports 250 mg/day of each of the two omega-3 PUFAs (docosahexaenoic acid (DHA) and eicosapentaenoic acid (EPA)) for adults [72]. Recently, an untargeted metabolomic mass spectrometry analysis reported reduced amounts of serum fatty acid metabolites in myopic subjects compared to non-myopic subjects; in particular, serum DHA levels were significantly lower in the myopic group [73]. Considering that DHA and EPA also have effects on hypoxia signaling [74,75] and myofibroblast transformation [76,77], researchers also believe that these effects of DHA and EPA are also considered to be related to the inhibition of myopia development.

Recent studies have shown that daily gavage of ω-3 PUFA (300 mg docosahexaenoic acid [DHA] plus 60 mg eicosapentaenoic acid [EPA]) significantly reduces the development of form deprivation myopia in guinea pigs and mice, as well as lens-induced myopia in guinea pigs [78]. Periocular injection of DHA also inhibited the progression of myopia in guinea pigs with form deprivation. Further experiments showed that DHA could increase choroidal blood flow and inhibit scleral hypoxia-induced myopia. In vitro experiments also demonstrated that treatment with DHA or EPA antagonized hypoxia-induced myofibroblast transdifferentiation in cultured human scleral fibroblasts.

### 7.2. PPARγ Agonists

Peroxisome proliferator-activated receptors (PPAR) isoforms are ligand-activated transcription factors that belong to the nuclear hormone receptor superfamily and have many various regulatory functions. The PPAR family includes three isoforms, PPARα, PPARβ and PPARγ, of which PPARγ is widely recognized as an important regulator of lipid metabolic processes [79]. Previous studies have shown that the development of myopia may also be related to PPARγ [80,81]. Moreover, previous studies in other tissues showed that PPARγ expression was affected by hypoxia. For example, activation of ERK1/2 and NF-κB by hypoxia can stimulate NOX4-mediated H_2_O_2_ production, thereby reducing PPARγ expression and activity in human pulmonary artery smooth muscle cells [82]. PPARγ has also been shown to be inhibited by hypoxic stress during adipocyte differentiation [83].

A recent study found that PPARγ activation and antagonism can affect the regulation of scleral hypoxia and collagen expression levels [84]. In a guinea pig model of myopia (FDM, induced by the occlusion of one eye and control of the other eye), periocular injection of the PPARγ agonist GW1929 in the occluded eye inhibited the decrease in choroidal thickness (ChT) and choroidal blood perfusion (ChBP), reduced HIFα expression, and upregulated collagen I expression. Thus, the progression of FDM was inhibited. Conversely, the PPARγ antagonist GW9662 could induce the development of myopia, upregulate the expression of HIF1a, and downregulate the expression of collagen I in the unshielded periocular area.

Together, these results suggest that the inverse relationship between changes in PPARγ and HIF-1α expression levels regulates collagen expression levels and controls the development of FDM. It also provides a new intervention method for the treatment of myopia by targeting scleral hypoxia. However, the specific regulatory mechanism between PPARγ and HIF-1α in myopia development needs further exploration.

### 7.3. Atropine

Atropine, a non-selective muscarinic receptor (cholinergic M-type receptor) blocker, has been clinically used for a long time. Historically, atropine was mainly used to treat vomiting, nausea, and bradycardia, and it was used in combination with other anesthetic drugs to avoid vagus nerve depression [85]. Over the past two decades, atropine has emerged as a preferred option to delay the progression of myopia or treat it [86]. The myopia control effect of atropine eye drops shows a concentration-dependent effect, and the control effect of high-concentration atropine eye drops on myopia can be as high as 60–96% [87,88]. However, high-concentration atropine eye drops have adverse reactions such as severe photophobia, a decrease in near vision, and rebound effects after drug withdrawal [89].

Continuous clinical studies both domestically and internationally have found that 0.01% (low-concentration) atropine not only exerts a myopia control effect [88,90,91,92] but also has high safety and fewer side effects. Notably, the rebound effect after drug withdrawal is less pronounced. Recent studies have found that atropine can prevent and control myopia through multiple pathways, either directly or indirectly, affecting the retina, choroid, or sclera [8]. Intravitreal injection of atropine has been shown to increase dopamine concentration in the chicken retina [93]. Atropine rapidly induced transient choroidal thickening and inhibited the development of myopia in a chicken LIM model [94]. Atropine also increases the thickness of the scleral fiber layer in the myopic eyes of chicken [95] and mouse [96] models.

Recent studies by proponents of the myopia scleral hypoxia theory suggest that atropine’s mechanism in preventing and treating myopia may also be related to choroidal blood flow and scleral hypoxia. Atropine significantly increases choroidal blood flow and inhibits axial length elongation and scleral hypoxia induced by myopia [43]. However, the molecular mechanisms by which atropine increases choroidal blood flow and regulates scleral hypoxia remain unclear and warrant further exploration.

### 7.4. Electroacupuncture

As one of the important treatment methods of traditional Chinese medicine, acupuncture is widely used in the prevention and treatment of angina [97], stroke [98], palpitations [99], coronary heart disease [100], and other diseases. Acupuncture is also increasingly used clinically to treat many ocular disorders, including dry eye syndrome [101], ptosis [102], oculomotor nerve palsy [103] and other ocular disorders [104], with promising results. As a further modified form of acupuncture and moxibustion, electroacupuncture (EA) is used to achieve therapeutic effects by generating electrical pulses through an external device. Previous studies have shown that electroacupuncture can be used as a clinical treatment technique to prevent ischemic diseases, and electroacupuncture preconditioning is an important protective mechanism for tolerance to ischemia [105,106,107,108].

Therefore, relevant researchers have proposed that electroacupuncture may improve scleral hypoxia by increasing choroidal blood flow, and that it plays a crucial role in the occurrence and development of myopia. In a study of electroacupuncture (EA) for LIM (lens-induced myopia) guinea pigs, it was found that EA could significantly reduce the diopter and axial length and significantly increase choroidal thickness and choroidal capillaries in myopic guinea pigs. Moreover, EA treatment increased the scleral collagen fiber diameter in myopic guinea pigs and significantly reduced the mRNA and protein expression levels of HIF-1α and MMP-2 in the sclera of myopic guinea pigs [109]. Another study also showed that EA improved retinal function and delayed the onset and progression of myopia by improving retinal blood flow, reducing retinal oxidative damage, inhibiting the RhoA/ROCK2 signaling pathway, and controlling extracellular matrix remodeling (increased expression of collagen type I and α-SMA) [110]. Moreover, researchers have found that electroacupuncture can affect the mitochondria-dependent apoptosis signaling pathway of ciliary muscle in LIM mice, thereby inhibiting the development of myopia [111].

In conclusion, these studies suggest that EA may delay the development of myopia through the pathway of choroidal blood flow-scleral hypoxia and may also contribute to a deeper understanding of the pathogenesis of myopia and provide a theoretical basis for optimizing acupuncture regimens and EA for the treatment of myopia.

### 7.5. Salidroside

Salidroside (SDS) as a phenolic compound is extracted from the tubers of Rhodiola plants and it has been implemented in various diseases due to its many significant biological activities, such as NAFLD/NASH [112], osteoporosis [113], and psoriasis [114].

Recently, researchers found that SDS had potential anti-hypoxia effects and could be a potential novel therapeutic agent against hypoxia injury. For instance, SDS has been proved to protect liver [115,116], renal [117], and pulmonary arteries [118] from hypoxia injury. Thus, researchers injected SDS into the FD eyes of guinea pigs for 4 weeks and found that higher-dose SDS (10 ug per eye) periocular injections significantly inhibited the development of myopia in FD guinea pigs with the inhibition of elongation of axial length and vitreous chamber depth [11]. And its ameliorative effects were associated with the downregulation of HIF-1α expression and the upregulation of COL1α1 expression in sclera [11]. Collectively, SDS as an anti-hypoxia drug has the potential to alleviate scleral hypoxia and improve myopia.

### 7.6. Formononetin

As an isoflavone, Formononetin (FORM) belongs to the phytoestrogen group and have a variety of physiological effects on health [119]. Especially in the medical area, FORM has been identified as a potential active compound to prevent or treat several diseases, such as obesity [120], neurodegenerative diseases [121,122], and cancer [123].

Also known as an anti-hypoxia drug [124], FORM can indeed decrease the protein expression of VEGF, HIF-1α, and PHD-2 in ARPE-19 cells under hypoxia and prevent hypoxia-induced retinal neovascularization. In addition, daily injection with FORM in FD eyes of guinea pigs for 4 weeks could significantly inhibit the elongation of axial length and the improvement mechanism of FORM was similar to SDS [11].

### 7.7. Prazosin

Prazosin (PZ) is commonly used as an oral vasodilator to treat hypertension and congestive heart failure [125]. Considering the advantageous effect of the increase in choroidal blood perfusion (ChBP) on scleral hypoxia in myopia, researchers injected PZ daily into the form deprivation eyes of guinea pigs and found that PZ significantly increased ChBP and inhibited the progression of myopia, with a decrease in axial length and an improvement in scleral hypoxia [43]. In conclusion, these results suggested that PZ could be an activator of ChBP to relieve scleral hypoxia and delay the development of myopia.

### 7.8. Crocetin

Crocetin (CC) is a bioactive metabolite produced in biological systems by the hydrolysis of crocin [126]. Modern pharmacological investigations revealed that CC could exert a protective effect on the heart, liver, and nervous system and improve symptoms of diabetes, atherosclerosis, and depressive disorder [126].

As a well-known vasodilator, CC has also been reported to enhance blood flow in various organs, especially in the eyes [127]. The proposed underlying mechanisms of CC have been examined in the context of ocular disease [128]. In a multicenter randomized, double-blind, placebo-controlled trial, CC was able to decrease axial length, increase choroidal thickness (ChT), and attenuate the progression of myopia in children [129]. Furthermore, these protective effects of CC were also identified in a mouse model of LIM [130]. Although there is no direct evidence of improvement in ChBF with crocetin, some researchers have suggested that CC could alter ChT by increasing ChBF, and this view needs further investigation [127]. In brief, CC has been shown to be a potential anti-myopia drug, but whether its influencing mechanism is relevant to scleral hypoxia needs more reliable research data.

### 7.9. Ginko Biloba

Numerous studies have investigated the biochemical characteristics of *Ginko biloba* (G. biloba) and proved that it has a considerable benefit to common medicine [131,132]. Recently, ophthalmic researchers shifted their attention to G. biloba and regard it as a promising candidate for mitigating myopia. In LIM mice, G. biloba extracts showed an ability to enhance ChT and ChBP and decrease axial length [133]. These findings suggest that G. biloba also has the potential to be an anti-myopia drug and improve myopia by alleviating scleral hypoxia.

### 7.10. Bunazosin Hydrochloride

Bunazosin hydrochloride (BH) is an alpha1-adrenergic blocker and has long been seen as an effective eye drop for treating glaucoma [134]. In LIM mice, BH eye drops administration significantly suppressed the myopic shift in refractive error, axial elongation, and scleral thinning and increased ChBP [135]. This experiment is a good illustration of the effect of BH on improving myopia, which involves choroid and sclera remodeling. But further studies are also needed to investigate the underlying mechanism of BH.

### 7.11. Light Exposure

Previous studies have shown that outdoor activities could effectively delay the development of myopia, and that their preventive effect is associated with more light exposure (the duration of exposure and light intensity) [136,137,138]. In order to investigate the beneficial effects of light exposure on myopia, researchers implemented light exposure in in vivo studies, and their results indicated that bright light could prevent the development of experimental myopia [139,140], with results of choroidal thickening [141,142] and ChBP increase [43]. This phenomenon could be explained by the activation of the light-stimulated dopaminergic amacrine cells of the retina [20,143]. Notably, different chromatic light variations have various effects on myopia [144,145].

Collectively, light exposure affected retinal DA, increased ChT and ChBP, and prevented myopia in animal studies. Human studies also showed a strong relationship between light exposure and myopia. However, investigating the underlying molecular mechanisms of light exposure improving myopia is required to be fully comprehended As mentioned above, whether light can also improve myopia by affecting scleral hypoxia through the choroid (ChT and ChBP) like other drugs needs further study.

## 8. Summary and Future

This review summarizes recent studies on scleral hypoxia in myopia and demonstrates the theory that the retina–choroid–sclera hierarchy affects the development of myopia (Figure 3). But it is worth noting that the theory of scleral hypoxia, especially the hierarchical regulation relationship, requires systematic and detailed studies. The latest research also emphasized that histone lactylation is involved in the role of scleral hypoxia in myopia and provided the specific induction mechanism of myopia caused by scleral hypoxia. But the effect of non-histone lactylation in scleral hypoxia needs further investigation.

We further summarize the treatments for the prevention or cure of myopia, which affected the retina–choroid–sclera hierarchy theory at different levels or to some extent (Figure 4). But future studies should be implemented to provide stronger evidence to prove the mechanism of this hierarchical theory in myopia. In addition, significant studies are needed to identify the function of the inhibitors (Aminoflavone, EZN-2968) of the HIF-1 pathway in myopia. Notably, it is worth noting that most studies on these interventions have focused on animal models, such as chickens and guinea pigs (with the exception of atropine), and further clinical trials are needed to confirm the efficacy of these treatments.

## Figures and Tables

**Figure 1 ijms-26-00332-f001:**
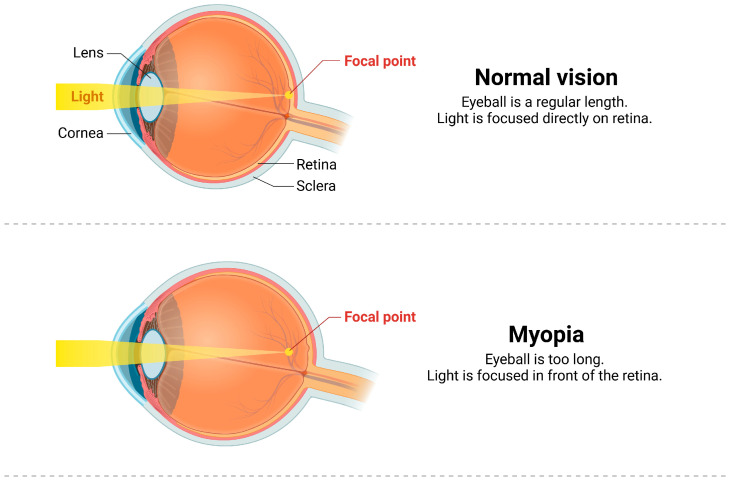
This figure illustrates the general ocular structural changes in myopia.

**Figure 2 ijms-26-00332-f002:**
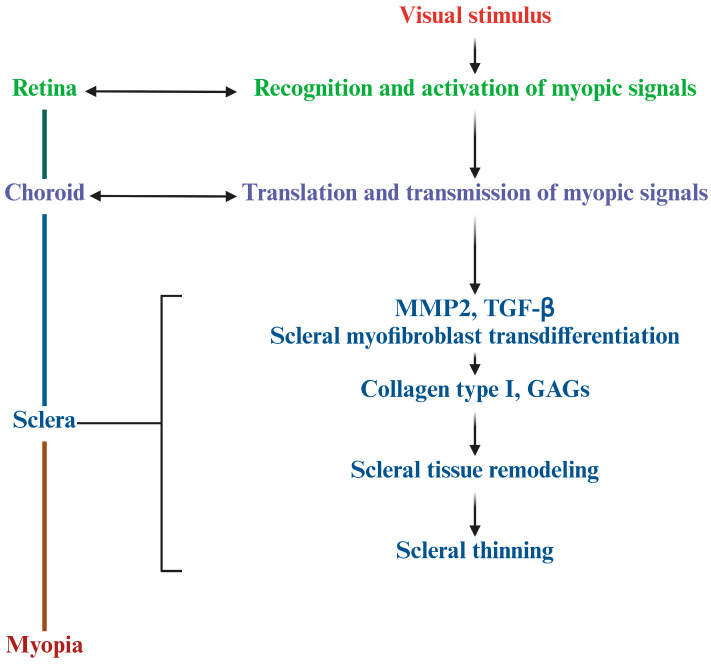
The mechanism of scleral remodeling during myopia.

**Figure 3 ijms-26-00332-f003:**
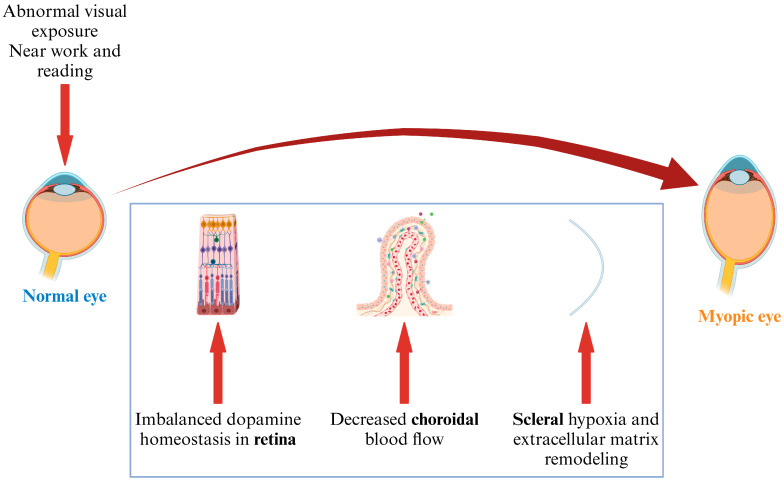
Theory of retina–choroid–sclera hierarchy during myopia.

**Figure 4 ijms-26-00332-f004:**
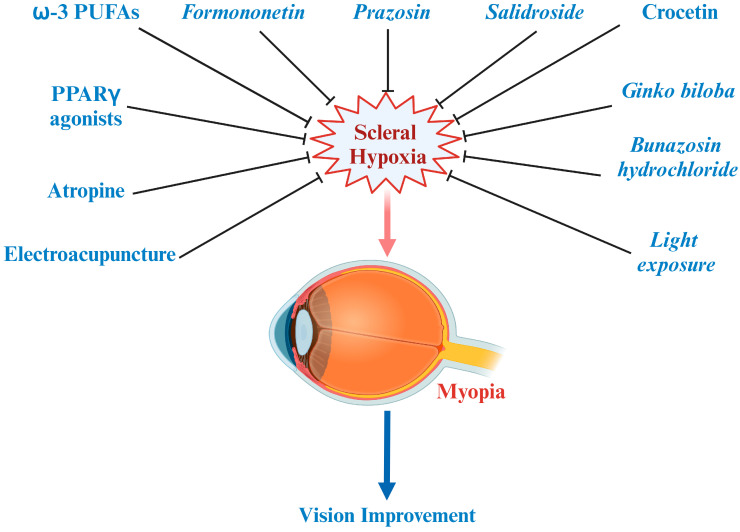
Treatments for the prevention and cure of myopia based on this theory.

**Table 1 ijms-26-00332-t001:** Main studies of hypoxia in myopia.

Authors	Findings
Wu H et al. [11]	Sclera scRNA-seq revealed that hypoxia signal (HIF-1α) is an important mechanism leading to scleral ECM remodeling in myopia.
Wu W et al. [12]	Sceral HIF-2α upregulation is also involved in myopia development.
Zhao F et al. [13]	GSA of GWAS data found that the HIF-1α signaling pathway was significantly enriched in patients with high myopia (diopter ≤ −10D).
Zhou X et al. [43,44]	Reduction in choroidal blood flow was the upstream inducing factor of scleral hypoxia.
Mathis U et al. [45]	Retinal dopamine controlled choroidal thickness.
Lin X et al. [46]	Lactate-induced histone lactylation contributed to the transdifferentiation of scleral fibroblasts into myofibroblasts and promoted myopia.

## Data Availability

Not applicable.

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
