# Peer review of "An Evidence-Based Narrative Review of Scleral Hypoxia Theory in Myopia: From Mechanisms to Treatments"

_ijms, 2025, doi:10.3390/ijms26010332_

Round 1
Reviewer 1 Report
Comments and Suggestions for Authors
Thank you to all the authors for putting this review together. Scleral hypoxia and myopia are interesting topics, and I really want to congratulate the authors on presenting this work. I would like to draw the authors' attention to a few sections.
1. It would be helpful if the author mentioned the type of review, such as narrative or systematic review; this would make it easier for the readers to understand.
2. if the authors have done a systematic review, then the methodology sections need to be added.
3. Defining pathological myopia in one or two sentences in the introduction section is needed.
4. line 68, add the word pathological in the sentence "predisposing factors of myopia."
5. Line 146: studies; adding a table with all the studies and their outcomes will help make the information easier and more clear.
Reviewer 2 Report
Comments and Suggestions for Authors
The general theme of the manuscript implies that hypoxia induces myopia.
However, causality does not appear proven by this review.
As in most cases of association, there is the possibility of confounders that are responsible for the association. Additionally, what evidence is that the causality is not reversed -- i.e. that myopia, or structural changes as part of myopia (axial elongation)/thinning of the sclera related to other factors, cause hypoxia rather than the other way around?
I find the general conclusions drawn in the paper to be far too strong for what has been presented.
Comments on the Quality of English LanguageGenerally good, though there are examples of English needing correction. For example, on the first page:
"However, the current epidemiological stud-39 ies of myopia support and do not support the view coexist. In recent years, the prevention 40 and control of myopia has become a global hot and difficult issue."
"In China, the incidence 41 of myopia in children and adolescents is high and younger, which seriously impacts the 42 physical and mental health of children."
Reviewer 3 Report
Comments and Suggestions for Authors
This is a well-written manuscript. However, several points are missing and should be added to increase the impact of the manuscript.
1) in Figure 2, the mechanism of scleral remodeling during myopia is depicted like this below. Retina (R) -> Choroid (Ch) -> Sclera (Sc) -> Myopia. R-Ch-Sc directions need more investigations. Arrow directions should be cautious. Therefore, it is recommended to use "<->".
2) HIF-target gene lists in the retina, sclera, and RPE/Choroid should be provided. Are there any differences? It will be very valuable to see some maps of activated genes by HIF modulation depending on the layer in the eye related to myopia.
3) Target drug or nutrient suggestions are quite limited. This should need more comprehensive candidates for myopia control.
4) Along with Comment 3, therapy (e.g. light or electric) could be added too.
5) Previously, myopia has been summarized as an ischemic eye condition: a review from the perspective of choroidal blood flow in a recent review paper (DOI: 10.3390/jcm13102777). It will be helpful to get more ideas of choroidal ischemia and blood flow in the eye with myopia and further myopia treatments to solve the comments (3 and 4).
6) Still, this concept of hypoxia and sclera with myopia is clearly needed to be studied using experimental models of myopia with sclera changes. Is it possible to summarize this aspect too?
Round 2
Reviewer 2 Report
Comments and Suggestions for Authors
Thank you for the significant revisions to your manuscript. Additional inclusion of experimental studies of HIF-knockdown help support the causality theory you propose/review in this article.
The addition of therapeutic targets was also very good.